# The relative importance of herbicide use for conservation tillage adoption by U.S. corn and soybean producers

Fengxia Dong[1], Laura Dodson[1], Rebecca Nemec Boehm[2]*, Cameron Douglass[3], Michelle Ranville[3‡], Ryan Olver[4‡]

1 United States Department of Agriculture, Economic Research Service, Washington, DC, United States of America, 2 United States Department of Agriculture, Office of the Chief Economist, Washington, DC, United States of America, 3 United States Department of Agriculture, Office of Pest Management Policy, Washington, DC, United States of America, 4 Tokyo International University, Tokyo, Japan

☯ These authors contributed equally to this work.
‡ MR and RO also contributed equally to this work.
* Rebecca.nemec@usda.gov

## Abstract

Herbicide use is widespread in agricultural production to control weeds prior to and after planting and to "burndown" weeds in the spring for conservation tillage. Whether conservation tillage adoption leads to higher herbicide usage has been a question of policy relevance for decades in the United States. Older U.S. studies using standard statistical and economic techniques have not consistently demonstrated higher herbicide usage levels among producers practicing conservation tillage, but these studies did not fully account for other practices, economic, or agronomic factors. To provide a more timely and comprehensive understanding of the importance of herbicides to conservation tillage, this study achieves two objectives with the most recent, nationally representative data from the US Department of Agriculture. First, it describes trends and compares conservation tillage and herbicide usage among field corn and soybean producers, similar to previous studies using standard economic techniques. Second, a Classification and Regression Tree (CART) model is employed—a novel methodology relative to previous studies that offers distinct advantages over traditional regression modeling—to evaluate the importance of herbicide use for conservation tillage adoption while accounting for other factors. Pairwise mean comparisons for field corn and soybeans indicated that herbicide usage pre-emergence was significantly higher with conservation tillage, but there was no consistent, significant differences in herbicide usage post-emergence. The CART analysis (with prediction accuracy ranging from 68–72%) also showed that pre-emergent use of glyphosate was the strongest predictor (with predicted probabilities from 0.83–0.86) of conservation tillage for field corn in 2016 and soybeans in 2018. Other factors such as the use of crop rotations, highly erodible land, region, and farm size were also strong predictors of conservation tillage. These findings highlight the importance and complexity of herbicide use in the adoption of conservation tillage for U. S. field corn and soybeans.

**Data Availability Statement:** The data used in this study consists of restricted microdata that cannot be shared publicly. However, information on how to access USDA ARMS data is available through

the USDA ERS website at: https://www.ers.usda.
gov/data-products/arms-farm-financial-and-crop-
production-practices/contact-us/.

**Funding:** This research was supported by the U.S.
Department of Agriculture, Economic Research
Service.

**Competing interests:** Authors report no competing
interests.

## 1. Introduction

A weed is practically defined as any plant growing where it is not wanted, and in the agricultural context, weeds can result in lower crop yields, lower crop quality, and increased production costs [1]. Weeds compete with agricultural crops for soil, nutrients, sunlight, and other resources, which is why their control is crucial to profitable and successful agricultural production [2, 3]. In the United States, herbicides were rapidly adopted from the 1950s to the 1970s and have become instrumental to the production of major commodities such as corn and soybeans, among others [4].

Broad-spectrum herbicides (i.e., those that control both grasses and broadleaved plants) such as 2,4-D, atrazine, and glyphosate are now in widespread use in U.S. commodity production [5–7]. These broad-spectrum herbicides are also used by producers practicing conservation tillage (hereafter defined collectively as no-till or mulch/reduced tillage) as a "burndown" tool to clear fields of weeds and other existing plants prior to or at the time of cash crop planting [8–12]. With conventional tillage, producers plow their fields and mechanically remove all plant residue from the top layer of soil instead of applying herbicide for this purpose.

While herbicides play a role in facilitating the implementation of conservation tillage, many agronomic and economic factors are also important. For example, producers may adopt conservation tillage to preserve soil moisture, prevent soil erosion, or reduce fuel, labor, and machinery costs [13, 14]. Additional factors include farm soil type, topsoil depth, local climatic conditions, availability and cost of necessary machinery, the age of existing tillage equipment and its sunk costs, as well as the price and availability of herbicides [15–18]. Reduced tillage also has numerous soil health benefits including additional carbon sequestration and reduced agricultural carbon emissions through lower fuel needs for pre-plant tillage, which may be another reason producers choose to adopt this practice [19–22].

How herbicides, in particular, factor into the decision to adopt conservation tillage has been a question of research and policy relevance for decades. Much of the research on this topic was motivated by the desire to inform policymakers about the difference in herbicide usage levels and intensity of use between conservation and conventional tillage. This interest grew particularly after federal lawmakers authorized conservation compliance policies in the 1990s, mandating less tillage for producers operating on highly erodible land [23]. Additionally, the increased availability and popularity of post-emergent (i.e., applied after emergence of the cash crop) herbicides and herbicide resistant crops became commercially available and more popular in U.S. crop production [24, 25] further motivated this research area. The body of prior research is relatively outdated and relied on standard economic and statistical techniques to evaluate the use of conservation tillage and herbicides in isolation, without explicitly accounting for other geographic, agronomic, and economic factors that would play a role in a producer's decision to adopt conservation tillage. Past research only qualitatively suggested that factors such as weather, soil type, and endemic weed issues were likely more influential to conservation tillage adoption and implementation than herbicide use [26]. Evaluating the relationship between herbicides and the decision to adopt conservation tillage in isolation without considering these other factors may lead to spurious or incorrect conclusions about the relationship between these two weed management practices.

Research from the late 1990s on field corn and soybean production did not consistently find higher rates or increased intensity of herbicide use among producers using conservation tillage compared to those using conventional tillage. A 1998 analysis by USDA's Economic Research Service and the Natural Resource Conservation Service using Cropping Practices Survey data collected between 1990 and 1995 among corn and soybean farmers found that herbicide application rates for field corn, soybean, and winter wheat producers were higher (11–

12% for corn and soybeans, and 68% for winter wheat) for those using conservation tillage systems versus conventional tillage, based on simple comparison of means across tillage type [26]. However, using data collected from USDA's Area Studies Survey of corn farmers in four watersheds in Indiana, Central Nebraska, Eastern Iowa, and Southern Illinois, Fuglie (1999) found that herbicide usage was not always higher on fields with conservation tillage systems, compared to those with conventional tillage, while controlling for some farm-level and geographic factors in a regression analysis framework. These results were consistent with earlier studies comparing herbicide usage across producers using conservation and conventional tillage [27–29].

More recently, data from the USDA Agricultural Resources Management Survey (ARMS) and the Conservation Technology Information Center (covering Illinois, Indiana, Iowa, Kansas, Michigan, Minnesota, Missouri, Nebraska, North Dakota, Ohio, South Dakota, and Wisconsin) were combined to evaluate the relationship between adoption of herbicide tolerant soybeans, conservation tillage, and "quality-adjusted herbicide use". This study found that adoption of the herbicide-resistant crop induced farmers to adopt conservation tillage while also decreasing "quality-adjusted herbicide use," a derived parameter the authors calculated based on annual per acre application rates, soil half-lives (a measure of the persistence of pesticides in soils), and "chronic toxicity scores", although this study only controlled for crop prices and no other farm-level, agronomic, or other economic factors [16]. Shortly thereafter a study using Kynetec's AgroTrak data from 1998 to 2011 covering all U.S. Crop Reporting Districts found significantly higher usage of herbicides for no-till producers compared to other producers, though the study only controlled for omitted variables using farm and year fixed-effects [30].

Studies on the cost of herbicides for producers using conventional versus conservation tillage also provide evidence on the difference in usage of herbicides across tillage practices. A case study by the Environmental Defense Fund, the National Corn Growers Association's Soil Health Partnership, and K-Coe Isom surveyed seven corn and soybean farmers in the Midwest in 2021, finding that herbicide costs were $6.00 per acre higher for those using conservation tillage versus conventional tillage [31]. A similar case-study approach of three corn, soybean, and wheat farmers found lower costs for chemicals, including herbicides, for those implementing conservation tillage versus conventional tillage [32]. It is important to note that these two studies were very small in scale, not representative of U.S. producers and thus results have relatively limited generalizability. Nonetheless, they are worth mentioning given that they provide newer data on the relationship between herbicide usage and conservation tillage.

To develop a timelier and more comprehensive understanding of the importance of herbicide usage to conservation tillage adoption, this study has two objectives. First, it documents the most recent temporal and geographic trends in 2,4-D, atrazine, and glyphosate usage, and the adoption of conservation tillage (which includes both no-till and mulch tillage) among U.S. field corn and soybean producers using the U.S Department of Agriculture's Agricultural Resource Management Survey (ARMS) data collected in 2012 and 2018 (for soy) and 2016 and 2021 (for field corn). The study also evaluates the importance of herbicide usage to conservation tillage using pairwise mean comparisons on three indicators of herbicide usage by tillage practice type (conventional, mulch till, no-till) for field corn (in 2016 and 2021) and soybeans (in 2012 and 2018). This study focuses on field corn and soybeans given their overall contribution to U.S. agricultural crop production (in terms of both value and total acres planted) and the relative abundance of USDA data on tillage practices and herbicide usage data for these crops. Atrazine and glyphosate in field corn, and 2,4-D and glyphosate in soybeans are the herbicides of focus, given the relatively high pre- and post-emergence usage of these herbicides in these two crops and pending regulatory changes for these herbicides under consideration by the U.S. Environmental Protection Agency [33].

Second, because conservation tillage adoption is driven by a number of factors, not just herbicide usage, a Classification and Regression Tree (CART) model is employed to analyze how the usage of these herbicides, along with other agronomic and economic factors, predicts the adoption of conservation tillage by U.S. field corn and soybean producers. Through the CART analysis, the relative importance of herbicide usage to conservation tillage among field corn and soybean producers is identified in the context of numerous other agronomic and economic factors. The CART model has not been employed to assess the importance of herbicide usage for conservation tillage previously. This methodology is advantageous because it captures complex, non-linear relationships and interactions between variables and is robust to outliers. Unlike traditional regression models, CART avoids issues such as endogeneity and multicollinearity, which can arise from including many control variables. Thus, CART provides a novel approach to enhance knowledge and understanding of the importance of herbicides to conservation tillage use.

## 2. Materials and methods

### 2.1. Data

This study uses data from the USDA Economic Research Service (ERS) and National Agricultural Statistics Service (NASS) Agricultural Resource Management Survey (ARMS) for field corn for 2016 and 2021 and soybeans for 2012 and 2018. ARMS is a nationally representative survey carried out in three phases that targets the minimum number of states that represent 90 percent of all U.S. field corn and soybean acreage, which allows for results that are nationally representative of U.S. field corn and soybean acreage. Phase 1 screens farms for eligibility based on the commodity they are producing in the survey year and whether they expect to or normally sell >$1,000 worth of agricultural products. Phase 2 asks detailed questions about quantity of inputs used, production practices, and production costs for a randomly selected field on their farm operation. Phase 3 collects farm-level information about farm demographic characteristics and finances. The states in the 2016 and 2021 survey for field corn include North Dakota, South Dakota, Kansas, Pennsylvania, Ohio, Indiana, Wisconsin, Kentucky, North Carolina, Illinois, Nebraska, Iowa, Michigan, Colorado, Minnesota, Missouri, Texas, and New York. States included in the 2012 and 2018 soybean survey include Tennessee, Ohio, Kentucky, North Carolina, Kansas, Arkansas, Nebraska, Michigan, Minnesota, Mississippi Wisconsin, Missouri, Louisiana, South Dakota, Indiana, Illinois, North Dakota, Iowa, and Virginia. For this analysis, data from Phase 2 for field corn (2016 and 2021) and soybeans (2012 and 2018) are used.

The main outcome variable of interest is whether the producer was implementing conventional tillage, mulch tillage, or no-till on their ARMS sample field during the survey year. The Soil Tillage Intensity Rating (STIR) is used to classify producers as implementing conventional tillage, mulch tillage, or no-till on their sample field. STIR, a measure developed by the USDA's Natural Resource Conservation Service (NRCS), quantifies soil disturbance of tillage practices based on the type of tillage equipment used, tillage depth, speed, and the percent of soil surface distributed, and can range in value from zero to 200, with higher values indicating greater tillage intensity [34]. Data for these calculations come from Table F in Phase 2 of the ARMS survey. Conventional tillage is defined as a combination of tillage management practices used between the harvest of the previous crop and the harvest of current crop that result in a STIR value >80. Mulch till is defined as tillage management practices used between the harvest of the previous and current crops that have a STIR value ≤80, where soil is tilled but soil disturbance is low. No-till is defined as the practice of not tilling between the harvest of the cash crop.

## 2.2. Methodology for examining current prevalence and trends in conservation tillage and herbicide usage

Using ARMS data for corn (2016 and 2021) and soybeans (2012 and 2018), national trends in conservation tillage and herbicide usage levels are described. After determining the tillage type used by each respondent (either conventional or conservation tillage), the share of respondents using these practices is calculated to evaluate their prevalence and trends between the two time periods for each crop, by ERS Resource Region (which depicts the geographic specialization of U.S. commodity production [35]) and by state for field corn and soybeans. As described in the ARMS documentation, weights to calculate these shares were applied because of the probability-based sampling procedure used to collect ARMS responses [36].

Similarly, trends in average herbicide usage indicators over the two time periods by tillage type used are evaluated. To compare herbicide usage by tillage type (i.e., conventional versus conservation tillage), comparisons of mean tests for three measures of herbicide usage were conducted for: (1) percent of crop acres treated with the herbicide (PCT), (2) pounds of herbicide applied per treated acre (lbs./acre), and (3) the frequency of herbicide applications per acre (no. applications/acre) (note that this variable has historically been called "acre treatments"). Each measure was examined by application timing, i.e., total for the survey year; before/at planting (i.e., pre-emergence to the cash crop); and after planting (i.e., post-emergence to the cash crop). Table D of the ARMS Phase 2 questionnaire asks producers if any herbicides were used on the sample field during the crop year, and if the answer is yes, producers are asked to list the herbicide name (using a list provided [36]), what form it takes (dry or liquid), whether the product was part of a tank mix (a mix of multiple herbicides or other pesticides), how much was applied, and the amount applied per application to the sample field. These responses are then matched to a list of each product's active ingredient (ai), which is then used to estimate the amount (in pounds) of each chemical per product used. Variables on a per acre basis were calculated using producers' reported field size in acres.

Percent of crop acres treated with the herbicide is the maximum acres treated with any product containing the herbicide divided by the sample field size in acres. Pounds of herbicide applied per treated acre is calculated as the total pounds of the chemical applied to the sample field divided by the maximum treated acres times the number of applications of the product with the maximum value for treated acres; this value is representative of the single application rate (lbs ai/acre) for a given herbicide. The per-acre herbicide application frequency (also known as acre treatments) is the maximum treated acres times the number of applications of the product with the maximum value for treated acres, divided by the sample field size.

## 2.3. Methodology for pairwise mean comparison tests of herbicide usage by tillage type

Pairwise mean comparisons were made separately for each crop (i.e., field corn and soybeans) and herbicide (i.e., 2,4-D, atrazine, and glyphosate) to identify whether usage was statistically different for conventional versus conservation tillage and for pre- versus post-emergence application timings. Prior to mean comparison tests, the data were evaluated to determine whether they were normally distributed, using the Shapiro-Wilk test. When normality assumptions were met, a t-test was used to determine if mean herbicide usage was statistically different between the two groups (i.e., conventional tillage versus conservation tillage). When normality assumptions were not met, a Wilcoxon signed-rank sum or Kruskal-Wallis test was used to compare herbicide usage across tillage types. All tests were conducted using the 'survey' package (version 4.2.1) in R [37, 38]. As with the calculations of shares for conservation

practices, these tests were conducted with survey weights due to the probability-based sampling procedure used to collect ARMS responses [36].

## 2.4. Methodology for Classification and Regression Trees (CART) analysis predicting conservation tillage

To understand the relationship between conservation tillage and herbicide usage, a Classification and Regression Trees (CART) analysis was employed. CART is an analytical technique that splits a full dataset into more homogeneous groups through a binary recursive partitioning process [39]. In doing this, it helps researchers understand the most important explanatory variables in a dataset, which can inform additional explanatory models or studies using causal inference techniques.

CART has been used extensively in epidemiology and medical settings since the mid-1980s to tease out which human behaviors or characteristics are most important for delivering medical interventions most effectively [40]. In these contexts, it is often the case that many human behaviors or characteristics confound each other, which makes the use of standard statistical methods to determine which factors are most important to intervention delivery, such as multivariate linear regression, challenging and less useful. Further, practitioners have indicated that the results from regression models are harder for clinicians to interpret and apply in the clinical setting [41]. In the context of understanding drivers of conservation tillage adoption in agricultural production and the relative importance of herbicide usage, CART is a useful tool given that farmer behavior and production decisions are related and interactive, often confounded by multiple related factors.

CART is a simplified machine learning approach which can evaluate the relationship between many more variables relative to regression analysis, which means the cost of deciding which variables to analyze or include in a model is relatively low. Variables included in the analysis also do not need to follow a particular distribution, making it an advantageous alternative to regression analysis when data are not parametric. Further, there are limited ways in which the CART model can be adjusted or modified, which means it is relatively easy to use. A tolerance threshold for the recursive partitioning process and an a priori minimum group size must be set, but otherwise, model parameters or options are very limited. Similar to regression analysis, the key inputs to the CART analysis are an outcome variable and explanatory variables. Typically, researchers will train the model with a larger subset of observations from a dataset and then test it with a smaller subset of the same dataset.

Results from the CART analysis are displayed in a tree composed of a hierarchical set of branches or nodes, collectively indicating the level of importance of each explanatory variable in predicting the outcome variable of interest (in this case, adoption of conservation tillage). The first explanatory variable to split the data, which creates the first set of nodes, is considered the most predictive of the outcome variable. The second most predictive explanatory variable will create the second set of nodes in the tree, and so on. Each split of the data is meant to create more and more homogeneous groups within the data. Tree depth (i.e., the number of hierarchical splits in the tree) can be restricted to ensure that final results are not overly complex to interpret, but this is a subjective decision left to the researcher. The value that the explanatory variable takes in predicting the value of the outcome variable is also displayed on the tree. CART trees could theoretically split the data into so many groups that each observation (in this case, the observations are respondents in ARMS) is in its own terminal node, but this would not provide meaningful information. The researcher can adjust model parameters or "prune" the tree to ensure results are interpretable [42]. Pruning parameters include the

minimum group size at terminal nodes, minimum number of observations in a node for further splits, maximum tree depth, cross-validation error, complexity parameter (CP), and prediction accuracy.

The primary outcome of interest in the CART analysis is whether a producer is implementing conservation tillage versus conventional tillage. The primary explanatory variables of interest are the three indicating the level of herbicide usage, as described earlier in the methods section. The full list of variables used in the analysis can be found in S1 Appendix. These variables were identified by beginning with the full set of responses to the Phase 2 portion of the ARMS survey and selectively excluding survey questions that did not have a known economic or agronomic relationship to conservation tillage or herbicide usage based on prior literature and/or expert feedback. The selection process was liberal, including any variable with an evidence-based or hypothetical relationship to herbicide use or usage. Through this process, 67 variables were identified to be included in the soybean CART model and 90 for field corn. All included variables are described in S1 Appendix, and each variable is identified based on the ARMS questionnaire survey question from which it was derived. Some minor data cleaning and recoding were necessary for some of these variables, but nothing was done that would change the interpretation of the survey questions as they are stated in the ARMS questionnaires.

CART analyses were run in R using the 'rpart' package (version 4.1.21) on the 2016 field corn responses (n = 1,995) and the 2018 soybean responses (n = 2,260) [43]. The model was not run on the 2021 field corn responses given that the response rate in 2021 (n = 968) dropped significantly relative to those of 2016 and prior years of the ARMS field corn survey. It is not clear why the response rate in 2021 dropped so dramatically, however, it may be attributable to survey distribution challenges that arose during the COVID-19 pandemic. This is an area that warrants exploration in future research.

Twenty percent of the ARMS survey responses were randomly selected for each survey year for field corn and soybeans to test the model. The remaining 80% of the survey responses were used as the training data (with results from the model runs on the training data reported in the results section). The minimum group size at the tree terminal nodes was 20, which is the default value in the 'rpart' package. The maximum tree depth was iteratively adjusted to maximize the prediction accuracy of the models. For field corn, maximum tree depth did not impact prediction accuracy, so the model was left unrestricted. For soybeans, the maximum tree depth was set to seven, which provided the highest prediction accuracy compared to leaving the maximum depth unrestricted. The cross-validation error rate was evaluated along with the complexity parameter to ensure high model accuracy.

## 2.5. Methodology for interpreting of results using ARMS responses

ARMS questions are designed to ask producers about their production practices on a single field within their larger operation. Respondents are asked to list all of their fields, no matter how they define those fields individually, in their responses to the survey. Then ARMS enumerators randomly select a field from each producer's list of reported fields and instruct respondents to answer questions about production practices for the selected field only. Consequently, the survey responses used for this analysis are for the random field selected by ARMS enumerators and do not represent the entirety of production decisions for each respondent. However, the ARMS survey sample population of producers is constructed so that it is nationally representative of the population of producers for a particular commodity. Therefore, results are interpreted accordingly.

## 3. Results

### 3.1. Current prevalence and trends in conservation tillage adoption and herbicide usage by field corn and soybean producers

In 2016, 41% of U.S. field corn producers were implementing conventional tillage on their fields, while 59% were implementing conservation tillage In 2021, 28% were using conventional tillage and 72% were using conservation tillage. This represents a 13-percentage point swing towards conservation tillage adoption amongst U.S. field corn producers between 2016 and 2021.

When parsed by ERS farm resource region [35], conservation tillage adoption in 2021 exceeded 90% in the Eastern Uplands and Prairie Gateway resource regions, which respectively saw 28 and 12 percentage point increases in adoption between 2016 and 2021 (Fig 1). In 2016, conservation tillage adoption was lowest in the Fruitful Rim resource region, but this region also saw the greatest increase in adoption (+33 points) between 2016 and 2021. All resource regions experienced an increase in conservation tillage adoption from 2016 to 2021 except for the Northern Great Plains. Data for the Mississippi Portal region for field corn were not available because the ARMS survey does not cover the states included in it (LA, MS, AR, and TN), even though a small amount of field corn production does occur there.

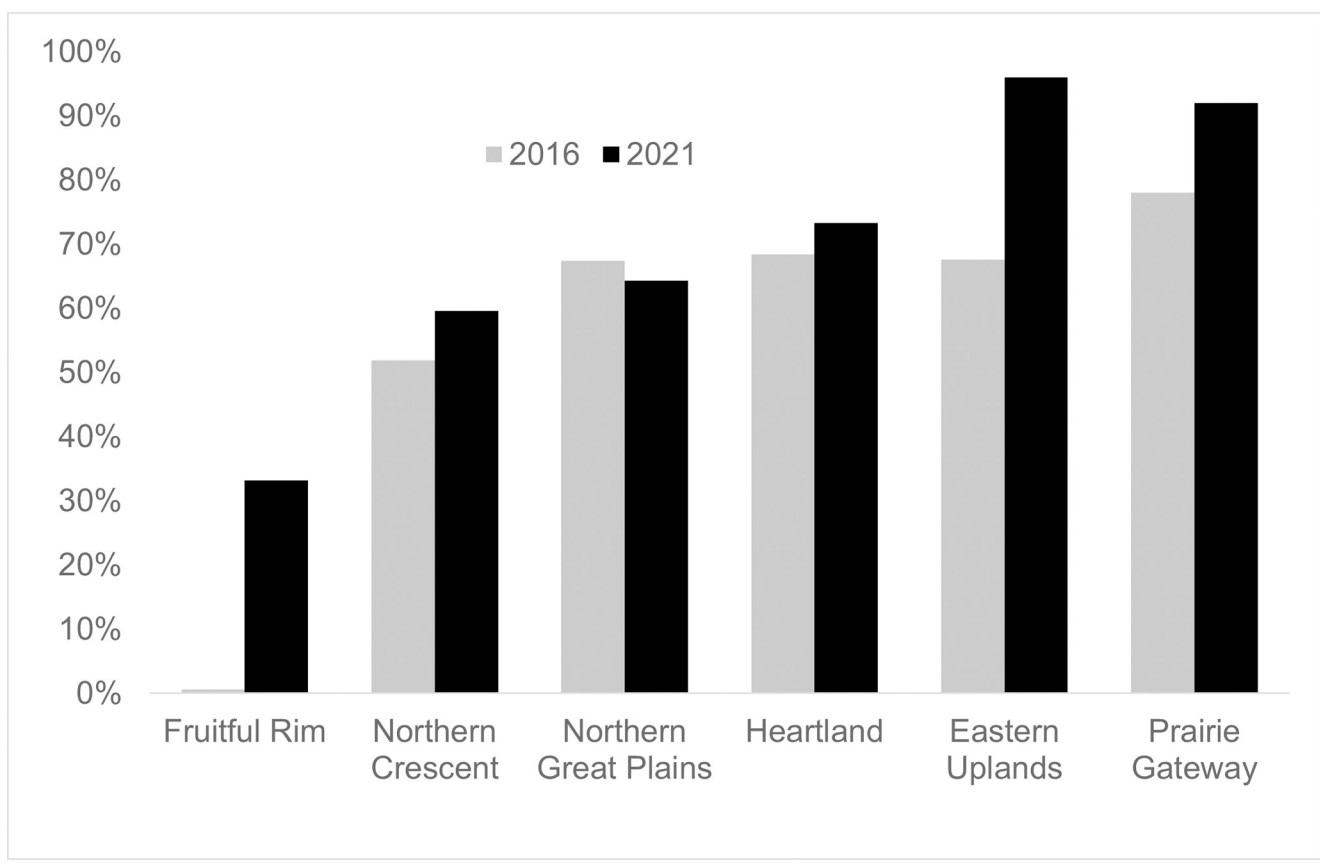

**Fig 1. For field corn, conservation tillage increased in all but one region between 2016 and 2021.** This chart shows the percentage of fields in the ARMS sample for field corn that reported using conservation tillage in 2016 (n = 1,995) and 2021 (n = 968). The only region that did not experience an increase in the rate of conservation tillage over this period was the Northern Great Plains. Data are not reported for the Mississippi Portal because the ARMS field corn does not survey states in this region (including LA, MS, AR, and TN).

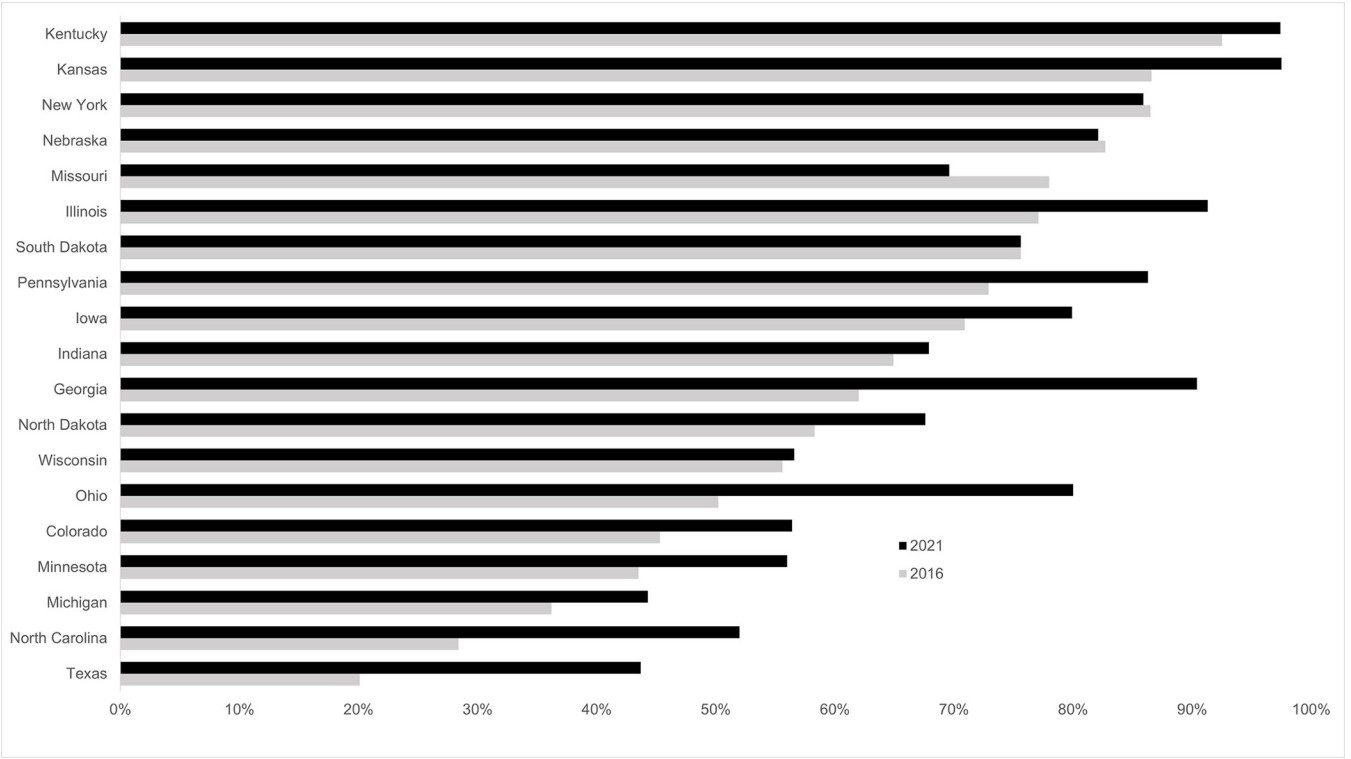

**Fig 2. For field corn, conservation tillage increased in all but three states from 2016 to 2021.** This chart shows the percentage of fields that reported using conservation tillage in 2016 (n = 1,995) and 2021 (n = 968) for the states included in the ARMS survey. The ARMS survey is only administered in states representing the vast majority of U.S. field corn production; therefore, not all states are represented in these results.

At the state level, conservation tillage adoption by field corn producers in 2021 exceeded 90% for Kansas (98%), Kentucky (97%), Illinois (91%) and Georgia (90%). In contrast, conservation tillage adoption was lowest for Texas and Michigan (44%) field corn producers (Fig 2). Increases in conservation tillage adoption between 2016 and 2021 were highest for Ohio (+ 30 percentage points), Georgia (+ 28 points), and North Carolina and Texas (+ 24 points). Nebraska and New York saw very minor declines (- 0.5 points) in conservation tillage adoption, while conservation tillage adoption in Missouri fell by almost 8.5 points; there were concomitant and proportional increases in the utilization of conventional tillage in these states.

In 2012, 25% of U.S. soybean producers were implementing conventional tillage, while 75% were implementing conservation tillage. This remained largely unchanged in 2018, with 22% implementing conventional tillage and 78% implementing conservation tillage. Nationally, there was only a 3-percentage point swing towards conservation tillage adoption amongst U.S. soybean producers between 2012 and 2018.

When parsed by ERS farm resource region, conservation tillage adoption in 2018 exceeded 90% in the Eastern Uplands, Southern Seaboard, and Prairie Gateway resource regions (Fig 3). However, the latter two regions saw minor declines (-2 to -5 percentage points) between 2012 and 2018, while the Eastern Uplands experienced only a minor increase of 7 points. Conservation tillage adoption was lowest in the Mississippi Portal and Northern Crescent regions, which both saw minor increases (+ 2 to +4 points) between 2012 and 2018. The Northern Great Plains region saw the greatest increase in adoption (+ 17 points), resulting in overall conservation tillage adoption by almost two thirds of soybean producers in the region by 2018.

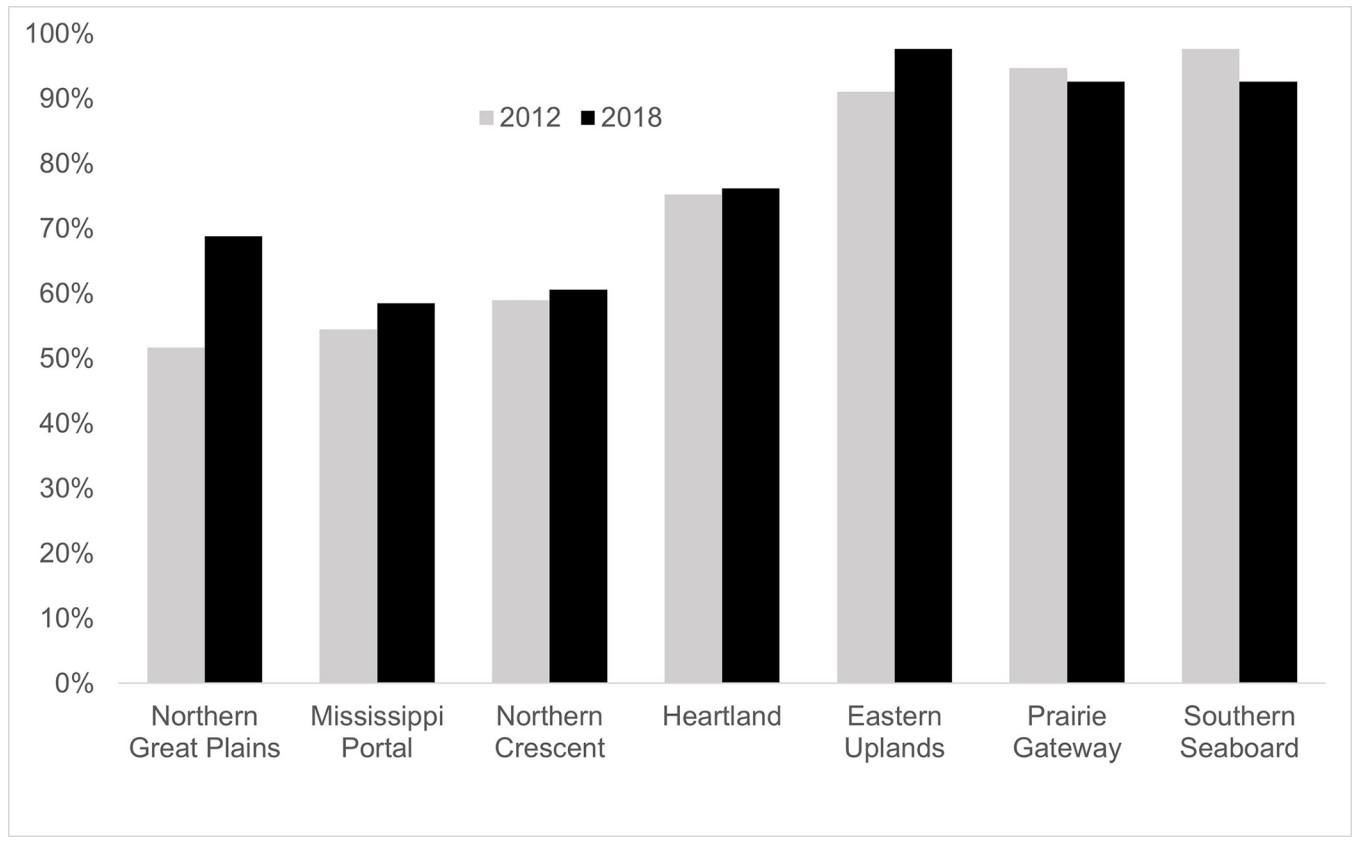

**Fig 3. For soybeans, conservation tillage increased in all but two regions from 2012 to 2018.** This chart shows the percentage of fields in the ARMS sample for soybeans that reported using conservation tillage in 2012 (n = 1,791) and 2018 (n = 2,260) by ERS Resource Region. No data are reported for the Fruitful Rim or Range and Basin regions because the ARMS soybean survey does not cover states in these regions even though some soybean production, albeit a small amount, does occur in them.

At the state level, conservation tillage adoption by soybean producers in 2018 exceeded 90% for Kentucky (100%), Tennessee (100%), Virginia (98%), Nebraska (96%), North Carolina and Missouri (91%). In contrast, conservation tillage adoption was lowest for Arkansas (29%), Minnesota (37%), and Louisiana (43%) soybean producers (Fig 4). Increases in conservation tillage adoption between 2012 and 2018 were highest for North Dakota (+ 20 percentage points), Missouri (+ 12 points), and Tennessee (+ 11 points). Conservation tillage adoption by soybean producers declined in multiple states between 2012 and 2018, specifically in Kansas (- 9 points), North Carolina and Ohio (- 7 points), Indiana (- 4 points), and Arkansas (- 1 points). There were concomitant and proportional increases in the utilization of conventional tillage in these states.

Overall, for both field corn and soybeans, glyphosate usage (measured as the percent of crop acres treated) was higher than that of atrazine (for field corn) or 2,4-D (for soybeans). In the surveyed years, on average, 77% of field corn acres received glyphosate treatments, compared with 62% and 65% of field corn acres receiving atrazine treatments in 2016 and 2021, respectively. For soybean acres, on average, 82–95% received glyphosate treatments compared to 15–19% of acres receiving 2,4-D treatments.

Average per acre single application rates for glyphosate were 0.91–1.00 lbs. ai/acre for both field corn and soybeans, but field corn acres tended to receive only one application per year on average, while soybean acres received between 1.30 and 1.60 applications per year, indicating

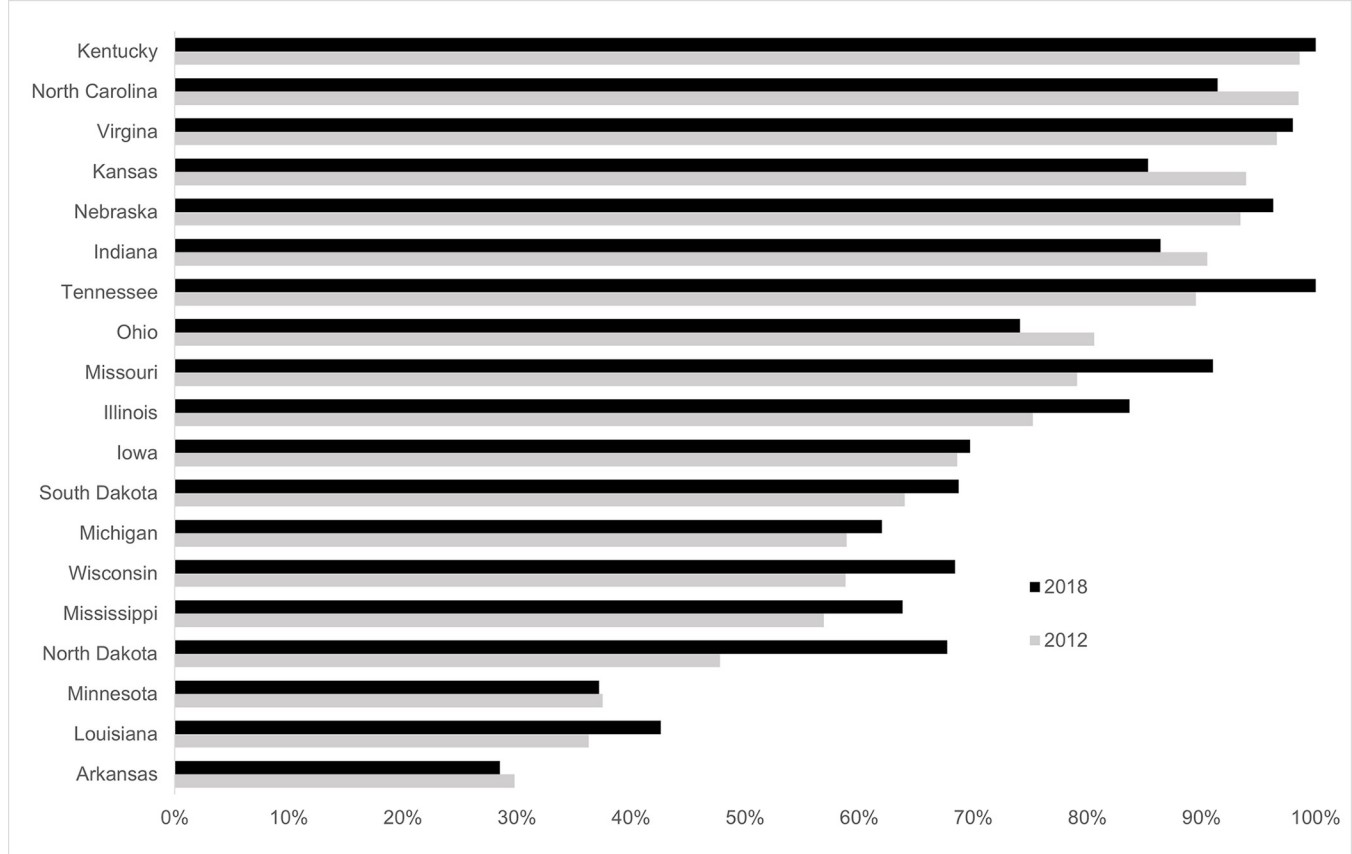

**Fig 4. For soybeans, conservation tillage adoption increased in most states from 2012 to 2018.** This chart shows the percentage of fields in the ARMS survey for soybeans that reported using conservation tillage in 2012 (n = 1,791) and 2018 (n = 2,260) among states included in the ARMS survey. The ARMS survey is only administered in states representing the vast majority of U.S. soybean production; therefore, not all states are represented in these results.

that individual application rates for glyphosate in soybeans tended to be lower than those in field corn (Table 1).

For corn, from 2016 to 2021, the average percent of acres treated with atrazine increased by 4.7%, while the average percent of acres treated with glyphosate decreased by 1.2%. Similar

**Table 1. Average usage of 2,4-D, atrazine, and glyphosate among U.S. corn (2016 and 2021) and soybean (2012 and 2018) producers.**

| Crop | Herbicide | Year | Average Percent of Crop Treated (%) | Average Single Application Rate (lbs ai/acre) | Average Annual Applications per Acre |
|------|-----------|------|-------------------------------------|-----------------------------------------------|--------------------------------------|
| Corn | atrazine | 2016 (n = 1,995) | 61.8 | 0.93 | 0.75 |
| | | 2021 (n = 968) | 64.7 | 0.88 | 0.84 |
| | glyphosate | 2016 (n = 1,995) | 77.4 | 0.93 | 1.02 |
| | | 2021 (n = 968) | 76.5 | 1.00 | 1.00 |
| Soy | glyphosate | 2012 (n = 1,791) | 95.0 | 0.96 | 1.61 |
| | | 2018 (n = 2,260) | 81.4 | 0.91 | 1.30 |
| | 2,4-D | 2012 (n = 1,791) | 15.0 | 0.54 | 0.15 |
| | | 2018 (n = 2,260) | 19.2 | 0.52 | 0.22 |

trends were observed for the average number of applications per acre for corn over the period (a 12% increase for atrazine and a 2% decrease for glyphosate). Average single application rates decreased for atrazine from 2016 to 2021 by 5.4%, but increased for glyphosate by 7.5%.

For soybeans, larger changes in herbicide usage were observed from 2012 to 2018. The average percent of acres treated with glyphosate decreased by 14.3%, while the average percent of acres treated with 2,4-D increased substantially (28.1%) from 2012 to 2018. The most dramatic changes in herbicide usage were in the average annual applications per acre for soybeans. Average annual applications of glyphosate decreased by 19.3%, but increased by 46.7% for 2,4-D. Average single application rates for glyphosate and 2,4-D decreased by 5.2% and 3.7%, respectively, over the same period.

## 3.2. Pairwise mean comparison of herbicide usage by field corn and soybean producers practicing conventional versus conservation tillage

The average percent of field corn acres treated with either atrazine or glyphosate under conservation tillage was 7 and 10 percentage points higher ($p \leq 0.01$) in 2016 relative to acres treated under conventional tillage. In 2021, there were no statistically significant differences in the percentage of acres treated with atrazine or glyphosate by tillage type among corn acres (S1 Table). There were no consistent differences in single application rates for atrazine or glyphosate by tillage type in 2016, though in 2021, glyphosate single application rates on corn acres under conventional tillage were 0.13 lbs. ai/acre lower ($p \leq 0.01$) than those under conservation tillage; application rates on conventional tillage acres fell relative to 2016, while rates on conservation tillage acres remained steady at 0.90–0.95 lbs. ai/acre. On conservation tillage acres, generally more than one application of glyphosate was made per acre in 2016 and 2021, significantly ($p \leq 0.001$ for 2016) more frequent than on conventional tillage acres, which received on average 0.83–0.89 applications per acre. Similarly, conservation tillage acres received more ($p \leq 0.01$) frequent applications (0.79–0.89) of atrazine than conventional tillage acres (0.65–0.73).

The same general trends were also observed with soybean acres (S2 Table), with the percent of acres treated with either 2,4-D or glyphosate, and under conservation tillage, being 15–16 and 7 percentage points higher ($p \leq 0.01$) for 2,4-D and glyphosate, respectively. There were no consistent statistically significant differences in single application rates for 2,4-D or glyphosate by tillage type in 2012 or 2018. On conservation tillage acres, glyphosate was applied 0.16–0.26 times more frequently than on conventional tillage acres, while 2,4-D applications were 0.17 times more frequently on conservation tillage acres ($p \leq 0.001$).

The apparent higher level of herbicide usage for both corn and soybean acres under conservation tillage (relative to acres using conventional tillage) is specifically for applications made pre-emergent to the crop (Fig 5, S1 and S2 Tables). In corn, while the percent of crop treated post-emergence was generally not different between conservation and conventional tillage, the percent of conservation tillage crop treated pre-emergence was 10–17 percentage points and 21–22 percentage points higher ($p \leq 0.001$) for atrazine and glyphosate, respectively. Likewise, the frequency of pre-emergence applications on conservation tillage acres was 0.24–0.25 and 0.13–0.19 times greater ($p \leq 0.001$) for atrazine and glyphosate, respectively. In soy, while the percent of crop treated post-emergence was unchanged (or, in the case of 2012, when conservation tillage acres treated with glyphosate fell by 5 percentage points ($p \leq 0.01$), the percent of conservation tillage acres treated pre-emergence was 13–15 percentage points and 29 percentage points higher ($p \leq 0.001$) for 2,4-D and glyphosate, respectively. Likewise, the frequency of pre-emergence applications on conservation tillage acres increased by 0.15–0.17 and 0.32–0.34 times ($p \leq 0.001$) for 2,4-D and glyphosate, respectively.

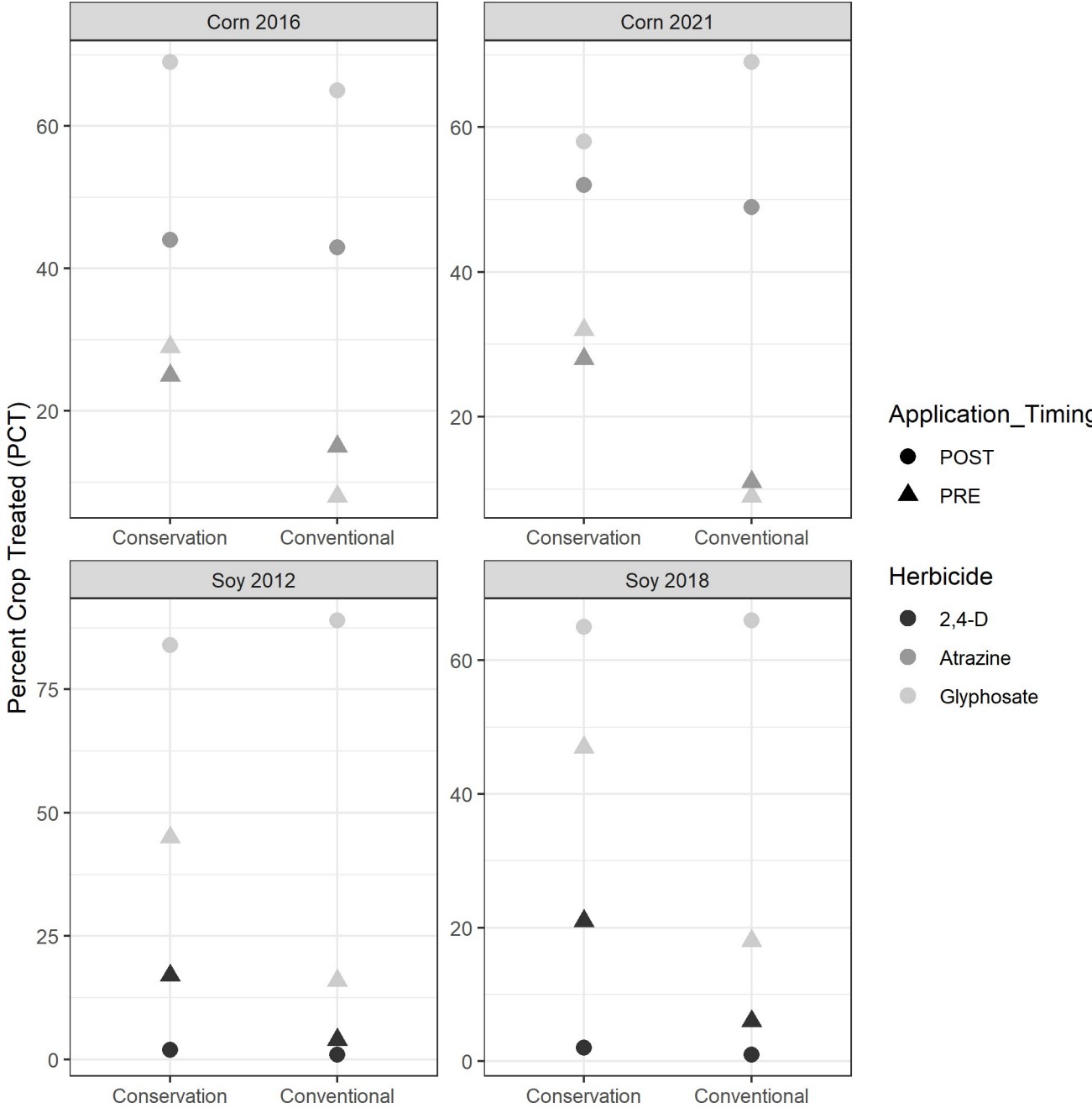

**Fig 5. Pre-emergent percent of crop treated is significantly higher for conservation tillage (both field corn and soybeans).** This chart shows the average percent of crop treated across field corn (2016, n = 1,995 and 2021, n = 968) and soybean (2012, n = 1,791and 2018, n = 2,260) fields for glyphosate, atrazine, and 2,4-D in the pre-emergent and post-emergent crop production phases.

## 3.3. CART analysis results

**3.3.1. Predictors of conservation tillage among field corn and soybean producers.** The results of the CART analysis predicting conservation tillage usage for field corn (using the 2016 ARMS responses) are presented in Fig 6; explanatory variables are described in detail in S1 Appendix. The prediction accuracy of this model is 68.5%, meaning that if the model were

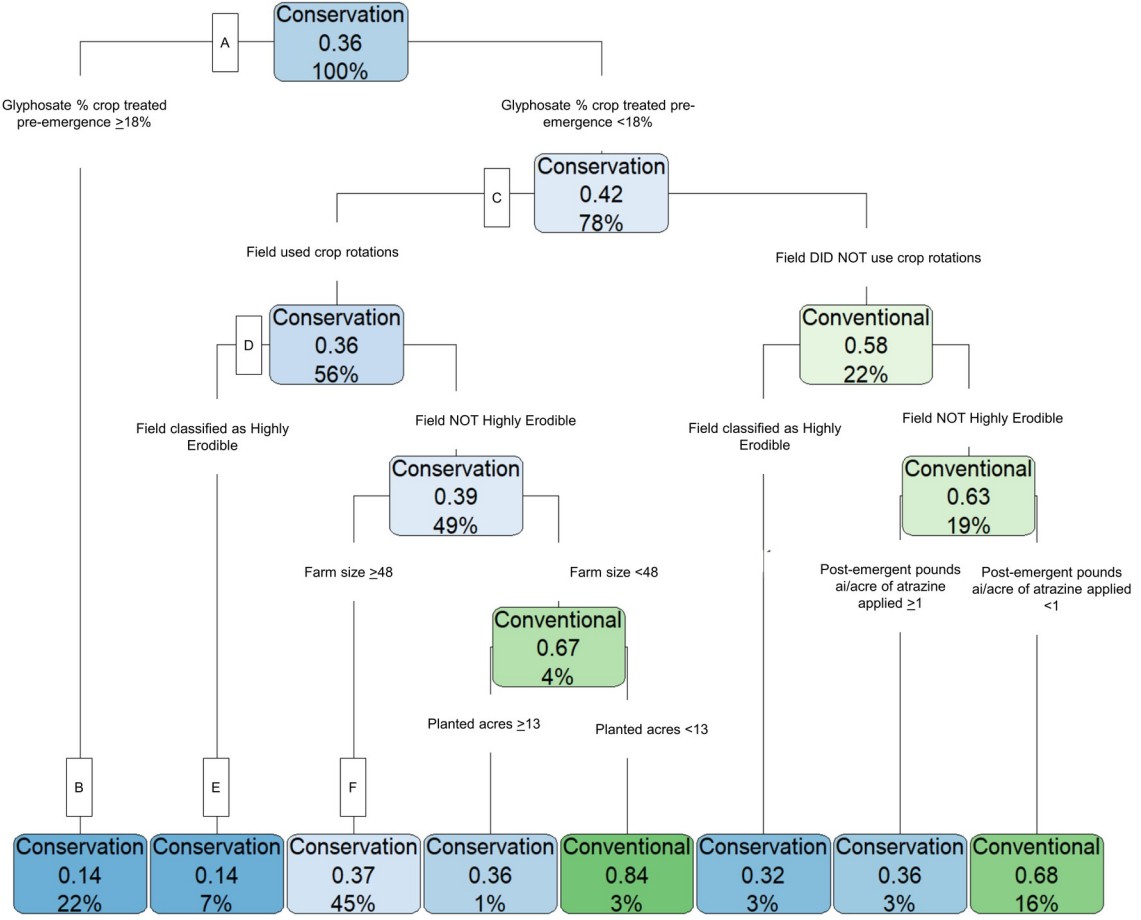

**Fig 6. For field corn, pre-emergent glyphosate percent of crop treated was the strongest predictor of conservation tillage.** This chart shows CART analysis results for 2016 field corn responses (n = 1,995). Blue shading indicates that the model predicts conservation tillage use for the subgroup, with a darker color indicating a higher predicted probability. The word "conservation" or "conventional" in each blue or green box shows the tillage type predicted for that subgroup of producers. The middle number, expressed as a decimal, represents the probability of conventional tillage adoption. For blue-shaded boxes where conservation tillage is predicted, the probability of conservation tillage is calculated as 1 minus the value in the box (e.g., at node B, the predicted probability of conservation tillage is 1–0.14 = 0.86). The last number, expressed as a percentage represents the share of the training sample that is categorized into that subgroup (e.g., at node B, 22% of respondents are in this subgroup).

applied to other data using the same explanatory variables, it would accurately predict a field or farm's use of conservation tillage 68.5% of the time. While a higher level of prediction accuracy is always preferred, other CART models of conservation tillage to compare prediction accuracy values are not available, making it unclear how to adjust the models to achieve improvements. This is an area that warrants future research.

The pre-emergent percent of crop acres treated with glyphosate most strongly predicts conservation tillage adoption, with the data splitting at 18% of acres treated (node A in Fig 6). This means that if ≥18% of a sample field's acres were treated with glyphosate, the model predicts that the field will be in conservation tillage with a predicted probability of 0.86 (node B). Twenty-two percent of the sample is categorized in this group and the remaining 78% of the sample require a more complex set of variables to explain conservation tillage adoption (beginning at node C). If the pre-emergent percent of crop acres treated with glyphosate is <18%, the model still predicts that a producer will use conservation tillage but at a lower rate (0.58 predicted probability).

Conditional on treating less than 18% of the field acres with glyphosate, the next most predictive explanatory variable was whether the producer's fields used a crop rotation, followed by whether the farmland was classified as highly erodible land (HEL) definition based on USDA Natural Resource Conservation Service standards (see S1 Appendix for full description). If crop rotations were used (i.e., if a crop other than corn was grown in a given field the previous year) the model's predicted probability of conservation tillage adoption rose from 0.58 to 0.64 and the model predicts that the field would be in conservation tillage (node D). Furthermore, if a field was in a crop rotation and also classified as HEL, then the predicted probability of conservation tillage adoption rises to 0.86 and the model predicts the field will be in conservation tillage (node E). Farm size and field acres planted play a minor mediating role in predicting conservation tillage.

The largest final group size included producers with a farm size $\geq$48 acres, on land not classified as HEL, using crop rotations, and those who treated less than 18% of their acres with glyphosate in the pre-emergent phase of production (node F). For these producers, conservation tillage is the predicted tillage type and the predicted probability is 0.63. For smaller farms and fields, conventional tillage is predicted, but this group represents only 3% of the training data sample.

Fig 7 presents results from the CART analysis of 2018 soybean producer responses. The prediction accuracy of this model is 72.2%, higher than that of the field corn model. Similar to the results for field corn, the most predictive explanatory variable for conservation tillage is the pre-emergent percent of field acres treated with glyphosate (node A). If the value is > 56%, the model predicts that the field will be in conservation tillage adoption (predicted probability of 0.90). The model predicted that 41% of the fields are categorized into this terminal node, which is the second largest group size among the terminal nodes. If the percent of acres treated with glyphosate in the pre-emergent phase is less than 56%, then conservation tillage use drivers differ by ERS resource region. For producers in the Eastern Uplands, Heartland, Northern Great Plains, Prairie Gateway, and Southern Seaboard resource regions, conservation tillage use is predicted (probability = 0.72) when producers apply less than 1.4 lbs glyphosate per acre per application (node B).

For producers who treat less than 56% of acres with glyphosate pre-emergent and are in the Northern Cresent or Mississippi Portal resource regions, conservation tillage is predicted if the field is less than 4.9 acres (predicted probability = 0.83). However, this group represents only 2% of the sample population.

## 4. Discussion

These analyses provide the latest information on the prevalence of conservation tillage adoption and herbicide usage among U.S. field corn and soybean producers. They offer a more comprehensive, timely, and novel evaluation of the significance of commonly used herbicides in relation to conservation tillage adoption. Importantly, the CART analysis circumvents the limitations of traditional statistical and econometric techniques, enabling a deeper understanding of the factors driving conservation tillage adoption among U.S. field corn and soybean producers.

### 4.1. Prevalence of conservation tillage and herbicide usage among U.S. corn and soybean producers

A key finding from this analysis is that a majority of U.S. field corn (72% in 2021) and soybean producers (75% in 2018) are using conservation tillage on their fields. This indicates a consistent upward trajectory in conservation tillage adoption nationally for both crops, with steady

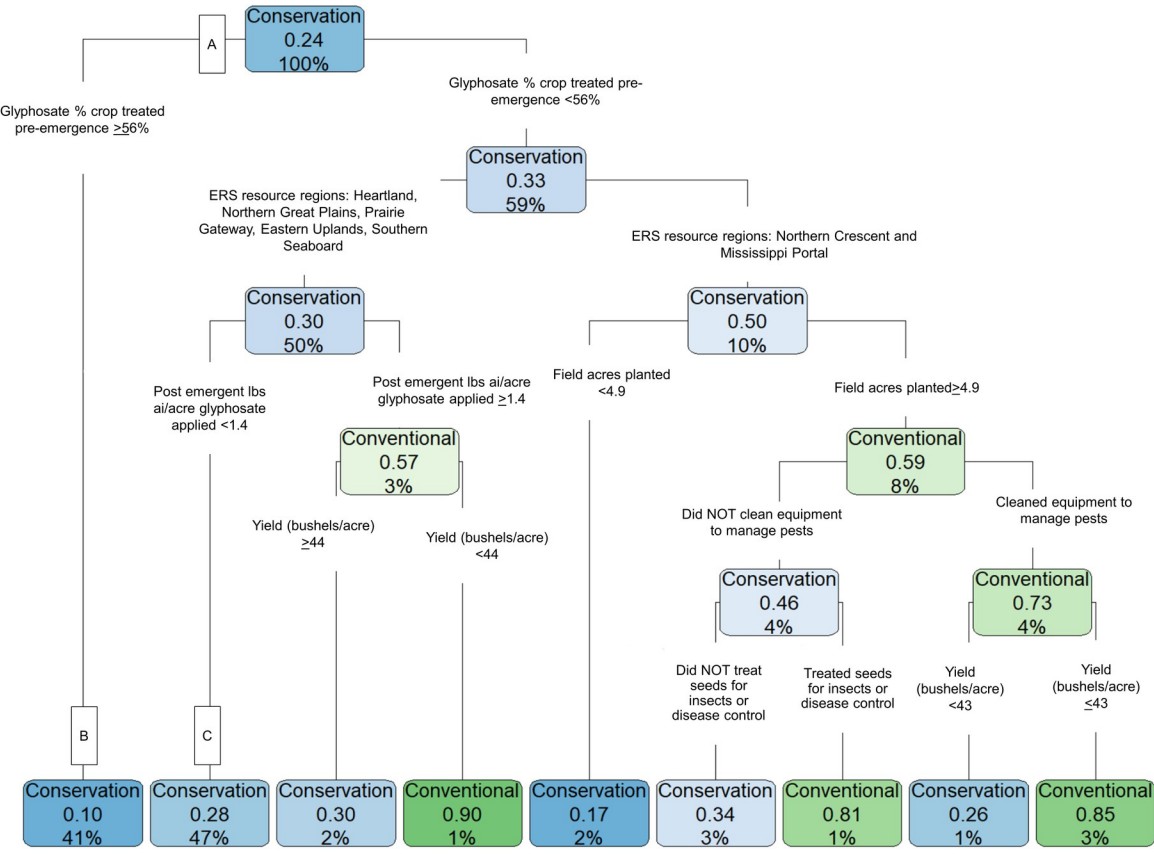

**Fig 7. For soybeans, pre-emergent glyphosate percent of crop treated was also the strongest predictor of conservation tillage.** This chart shows CART analysis results for 2018 soybean responses (n = 2,260). Blue shading indicates that the model predicts conservation tillage use for the subgroup, with a darker color indicating a higher predicted probability. The word "conservation" or "conventional" in each blue or green box indicates which tillage type the model predicts for that subgroup of producers. The middle number that is expressed as a decimal indicates the probability of conventional tillage adoption. For boxes shaded in blue, where conservation tillage is predicted, the predicted probability of conservation tillage is calculated as 1 minus the value in the box (e.g., at node B, the predicted probability of conservation tillage is 1–0.1 = 0.90). The last number in the boxes, expressed as percentages, indicate the share of the training sample categorized into that subgroup (e.g., at node B, the model categorizes 41% of respondents into this subgroup).

rates for soybeans. These trends align with those from 2010 to 2016 for field corn and from 2006 to 2012 for soybeans [13], suggesting no significant changes in the trajectory of conservation tillage adoption in more recent years. While national rates were generally higher or steady over the study periods for each respective crop, and most regions and states experienced increases in conservation tillage, some states still saw declines in adoption. For field corn, Missouri experienced an 11% decline, while Nebraska, New York, and South Dakota exhibited nearly flat trends. For soybeans, Indiana, Kansas, North Carolina, and Ohio reported declines in adoption ranging from 4% to 9%. Although this study did not investigate the drivers behind these temporal trends, future research should explore them, especially to understand dis-adoption, given the climate, production, and environmental benefits that conservation tillage offers.

Tracking the adoption of conservation tillage is crucial for several reasons. In this study, the primary aim was to understand its prevalence and trends as a foundation for assessing the importance of herbicide usage in conservation tillage adoption. However, it is also important to understand adoption rates, as changes in tillage practices affect producer input use and yield significant environmental and climate benefits. One recent study emphasized the importance

of tracking more localized trends in the adoption and disadoption of conservation tillage. A shift to conventional tillage can adversely affect soil carbon sequestration (which requires long-term use of conservation tillage) and influence emerging voluntary carbon markets and participation among U.S. farmers [44]. These findings support the idea of tracking conservation tillage adoption rates not just at the national level, but also at more granular geographic scales for multiple reasons, especially given overall increasing adoption rates nationally, but lower rates in some states and regions.

Prior to this study, the latest information describing national prevalence and trends in conservation tillage adoption came from a 2018 study, nearly 6 years ago, which included data for field corn through 2016 and for soybeans through 2012 at national, state and regional levels [13].The updated analysis of national, regional, and state-level prevalence and trends in conservation tillage adoption provides essential information that policymakers can use to continue to track conservation tillage adoption in U.S. field corn and soybean production. As of the latest available sampled years (i.e., 2018 for soybean and 2021 for field corn), the results indicate that roughly three quarters of U.S. acres of these crops were using conservation tillage (either mulch till or no-till), indicating that understanding the importance of herbicides to this practice is crucial given how widespread it is.

Trends in herbicide usage (from Table 1) indicate a decrease in all three indicators of glyphosate usage (average percent of crop treated, average single application rate, and average annual applications per acre) from 2012 to 2018 for soybeans, and for U.S. field corn from 2016 to 2021 for two of these indicators (average percent of crop treated and average annual number of applications per acre). Conversely, there were increases in the average percent of crop treated and average annual applications per acre for atrazine for U.S. field corn and 2,4-D for soybeans. No prior study presents such recent data on herbicide usage for a nationally representative sample of U.S. field corn and soybean producers, so a direct comparison of these results to other analyses is not possible. Despite this, the results suggest an apparent substitution pattern between glyphosate and atrazine (for field corn) and 2,4-D (for soybeans) over the study period. This may be due to increasing weed resistance to glyphosate, which has been documented in prior studies [45]. Another factor worth investigating is whether the expansion of herbicide-tolerant biotechnology traits has contributed to shifting herbicide usage. However, the rise of glyphosate-resistant weeds is the most likely driver. Future research is needed to gain a better understanding of these dynamics.

## 4.2. Comparison of pairwise mean comparison tests with previous studies

The pairwise mean comparison analysis found that field corn and soybean producers practicing conservation tillage had higher average percent crop acres treated and more frequent annual applications per acre pre-emergence compared to those practicing conventional tillage. However, there was no consistent difference in annual pounds of active ingredient per acre applied pre-emergence. Specifically, these results imply that producers practicing conservation tillage are shifting the timing of their herbicide applications to the pre-emergence timing window, but the application rates for these herbicides do not significantly differ between fields under conservation and conventional tillage. It is important to note that the calculation of the annual applications per acre indicator accounts for all corn and soybean field acres, which may introduce a potential downward bias, particularly for herbicides like 2,4-D that are typically used on a relatively small percentage of overall planted soybean acres.

Previous studies primarily examined herbicide usage in aggregate over the entire growing season, making these results, which disaggregate usage into pre-emergence and post-emergence periods, not directly comparable. However, this approach provides a more accurate

representation and comparison of herbicide usage across different tillage types. Furthermore, most prior studies were conducted over two decades ago, largely before the widespread commercialization of genetically-engineered, glyphosate-tolerant corn and soybean seed varieties [46, 47]. This updated analysis provides a more current and nuanced understanding of the potential impact of herbicide-resistant field corn and soybean varieties on the relationship between herbicide usage and conservation tillage adoption.

Weeds pose a major threat to productive agricultural systems, and effective direct weed management tools available to producers are arguably limited to herbicides, tillage, and cropping practices (e.g., rotations) [48]. Removing any one of these three tools increases reliance on the others, so producers practicing conservation tillage (especially no-till) largely depend on herbicides for weed control [8, 48, 49]. As herbicide-tolerant field corn and soybean seed varieties have become commonplace, producers have become accustomed to relying almost exclusively on herbicides such as 2,4-D, dicamba, or glyphosate for post-emergence weed control [11, 25, 49]. An axiom of modern farming in the U.S. is the importance of seeding into a field as free of weeds as possible, and pre-plant weed control is achieved either through tillage or, for producers using conservation tillage, with herbicides [49]. Therefore, these findings–that producers practicing conservation tillage treat a higher percentage of acreage with herbicides pre-emergence than those practicing conventional tillage–intuitively reflect the replacement of tillage with chemical weed control for the preparation of a weed-free seed bed.

## 4.3. CART model results

The CART analysis provides a more comprehensive picture of the relationship between herbicide usage and conservation practice adoption in the context of other production practices. It goes beyond this study's pairwise mean comparison and prior studies that compare mean usage of herbicides across different tillage types by accounting for many different agronomic, geographic, and other factors that may also play a role in determining a producer's decision to adopt conservation tillage.

The key finding of this analysis is that a higher average pre-emergent percent of acres treated with glyphosate is the most predictive factor of conservation tillage in the CART analysis. However, other factors, such as crop rotations, whether the land has been designated as highly erodible, region, and farm size were also strong predictors of conservation tillage. These results highlight the complex nature of weed management in many agricultural systems, and the inherent trade-offs therein. They point to the important role that crop rotations can have in aiding weed management by allowing producers to rotate herbicides and cultural weed management practices (e.g., row spacing) alongside crops, and by reducing the chances that a single weed species becomes unmanageable [9, 11]. There is growing evidence that the use of crop rotations or other cultural weed management practices, such as cover cropping, in addition to conservation tillage, further optimizes weed management programs and allows producers to reduce herbicide usage [8, 9, 49].

The CART analysis not only provides new information on how herbicide usage differs between conventional and conservation tillage, but also provides a ranking of the importance of herbicide usage in the context of many other production practices, farm size, and the geographic region and state of production. Results indicate that the percent of crop treated with glyphosate is central to the adoption of conservation tillage for both field corn and soybeans. However, other factors such as whether the field is in certain production regions or whether the field is considered highly erodible are also key predictors of conservation tillage adoption. These findings are intuitive because conservation tillage helps to preserve soil health, and if fields are classified as highly erodible, a producer is required to have a conservation plan that

includes the use of reduced tillage practices [50, 51]. Furthermore, soils vary geographically, so it is expected to see variation in conservation tillage adoption across regions. Additional studies are also needed to evaluate why larger farms, conditional on other factors (as illustrated in Fig 6), would be more likely to adopt conservation tillage.

Results from this study also demonstrate the potential for using CART to better understand decision making in agricultural production. There are limited examples of CART models being applied in the agricultural setting; one example is a recent working paper [52] that evaluates the relationship between conservation practice adoption and crop insurance use. CART models could, however, become a more common modeling tool used because they can be applied to better understand farmer decision making when many factors confound one another. CART models also have advantages over multiple linear regression in that they require fewer assumptions (i.e., data do not need to meet parametric assumptions), and variables that would typically be considered collinear can all be included. Moving forward, CART models could be used to explore how farm characteristics, production practices or other agronomic or economics factors are related to the adoption of cover crops or other conservation practices with environmental or climate benefits. Additionally, CART models can be used in preparation for causal inference studies because model results can help identify variables that confound the causal relationship between key explanatory and outcome variables. Few previous studies have evaluated the causal relationship between policy changes restricting or otherwise impacting herbicide usage and conservation tillage or other production practices, and the results of the CART models could help inform such future studies.

From a policy perspective, the information provided here is particularly important in the context of the recent infusion of funds into agricultural conservation programs in the Inflation Reduction Act, which aims to increase assistance to producers for adopting practices such as conservation tillage and no-till [53]. In addition, this analysis of conservation tillage adoption in the context of herbicide usage is especially timely, given the label mitigation changes for 2,4-D, atrazine, and glyphosate being considered by the EPA, both as part of routine registration review and the EPA's accelerated efforts to meet its consultation obligations under the Endangered Species Act [54, 55]. To fulfill its pesticide registration obligations under the Federal Insecticide, Rodenticide, and Fungicide Act (FIFRA), the EPA is considering changes to allowable uses on these herbicide labels that could directly constrain producers' use of herbicides to facilitate conservation tillage, for example, by discouraging the pre-emergence use of atrazine [33]. Under the ESA, the EPA is required by law to consult with the U.S. Fish and Wildlife Service (FWS) and the National Marine Fisheries Services (NMFS) to ensure that pesticide registrations do not imperil threatened and endangered species or their critical habitats [56]. One of the approaches the EPA has proposed to mitigate off-field risks to threatened and endangered species is by requiring producers to utilize on- and off-field practices that mitigate runoff and soil erosion, which includes conservation tillage [54]. The data in this study on the prevalence of conservation tillage amongst U.S. field corn and soybean producers illustrates both the potential magnitude of impact of these changes and the complexity of the drivers of conservation tillage practices.

### 4.4. Study limitations

In theory, the 2022 Census of Agriculture would provide the most up-to-date information on the prevalence of tillage practices. It was released in February 2024 and contains information on the adoption of conservation tillage among corn and soybean producers. However, this commodity-specific data on conservation tillage adoption can only be accessed through the restricted-use Census of Agriculture microdata or by requesting a special tabulation from the

National Agricultural Statistics Service; the authors were unable to acquire these data prior to publication to make comparisons to these results. Additionally, the Census of Agriculture asks producers whether they are adopting conservation, no-till, or conventional tillage and does not specifically calculate a STIR rating as ARMS does. These self-reported measures of tillage used in the Census of Agriculture are subject to more uncertainty because there is a wide variety of tillage practices within each type. STIR is a more objective measure that is preferable to use where possible. Consequently, while more updated data are available on the prevalence of conservation tillage among field corn and soybean producers in the U.S., these results represent the latest and most accurate information on conservation tillage adoption that currently exists in the literature.

## 5. Conclusions

This study provides evidence that conservation tillage adoption amongst U.S. field corn and soybean producers has continued to increase, with roughly three quarters of producers practicing conservation tillage as of 2018 (soybean) and 2021 (field corn). However, attention should be drawn to the geographic variability in adoption that may indicate dis-adoption dynamics. Pairwise mean comparisons indicate that producers using conservation tillage treat more acres with atrazine and glyphosate for field corn, and with 2,4-D and glyphosate for soybeans prior to planting than those using conventional tillage, but overall use is not significantly different between tillage types (see S1 and S2 Tables). CART models (with prediction accuracy ranging from 68–72%) confirmed that the pre-emergent percent of acres treated with glyphosate was the most important predictor of conservation tillage for field corn and soybean (predicted probabilities from 0.83 to 0.86; see Figs 6 and 7). Therefore, any label changes that would directly impact producer's ability to apply herbicides pre-emergent could impair continued practice of conservation tillage. CART model results pointed to the importance of crop rotations and other cultural weed management practices in facilitating conservation tillage, highlighting the role that Integrated Weed Management practices can have in optimizing weed control to support conservation practices.

## Supporting information

**S1 Table. Average usage of glyphosate and atrazine among U.S. field corn (2016 and 2021) producers practicing conventional versus conservation tillage.**
(DOCX)

**S2 Table. Average usage of glyphosate and 2,4-D among U.S. soybean (2012 and 2018) producers practicing conventional versus conservation tillage.**
(DOCX)

**S1 Appendix. Variables included in the CART model from the ARMS Phase 2 surveys.**
(DOCX)

## Acknowledgments

The authors would like to thank reviewers at the U.S. Department of Agriculture Economic Research Service and the Office of the Chief Economist for their feedback and suggestions that greatly improved the quality of this research article.

   **Disclaimer:** The findings and conclusions in this publication are those of the authors and should not be construed to represent any official USDA or U.S. Government determination or policy.

## Author Contributions

**Conceptualization:** Fengxia Dong, Laura Dodson, Rebecca Nemec Boehm, Cameron Douglass, Michelle Ranville, Ryan Olver.

**Data curation:** Fengxia Dong, Laura Dodson.

**Formal analysis:** Fengxia Dong, Laura Dodson, Rebecca Nemec Boehm, Cameron Douglass.

**Investigation:** Fengxia Dong.

**Methodology:** Fengxia Dong, Laura Dodson, Rebecca Nemec Boehm, Cameron Douglass, Michelle Ranville, Ryan Olver.

**Project administration:** Rebecca Nemec Boehm.

**Resources:** Rebecca Nemec Boehm.

**Visualization:** Rebecca Nemec Boehm.

**Writing – original draft:** Rebecca Nemec Boehm, Cameron Douglass.

**Writing – review & editing:** Fengxia Dong, Laura Dodson, Rebecca Nemec Boehm, Cameron Douglass, Michelle Ranville, Ryan Olver.

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
