## [Decision Letter · Decision Letter 0]

31 Jul 2024

PONE-D-24-09492The Relative Importance of Herbicide Use for Conservation Tillage Adoption by U.S. Corn and Soybean ProducersPLOS ONE

Dear Dr. Nemec Boehm,

Thank you for submitting your manuscript to PLOS ONE. After careful consideration, we feel that it has merit but does not fully meet PLOS ONE’s publication criteria as it currently stands. Therefore, we invite you to submit a revised version of the manuscript that addresses the points raised during the review process.

We look forward to receiving your revised manuscript.

Kind regards,

Dafeng Hui, Ph.D.

Academic Editor

PLOS ONE

Journal Requirements:

**Additional Editor Comments:**

Comments from Associate Editor:

This manuscript has now been reviewed by two reviewers. Both of them raised some concerns on the current version of the manuscript, and provided some suggestions. I briefly reviewed the manuscript and concur with the reviewer on most of their suggestions. Reviewer #1 recommended many references. While some of these are relevant to the work in the manuscript, there is no requirement to include any of these citations in their revised manuscript. Reviewer #2 provided some constructive suggestions for authors to improve manuscript. But item 3, whether using the first person in the technical writing process depends on the writing style. There is a trend of using the first person in scientific writing. I noticed that the manuscript was submitted a few months ago, but the delay was due to changes in AE and difficulty to find adequate reviewers. Sorry about this. Based on the reviewer's comments, my decision is Major revision.

Reviewers' comments:

Reviewer's Responses to Questions

**Comments to the Author**

1. Is the manuscript technically sound, and do the data support the conclusions?

Reviewer #1: Yes

Reviewer #2: Partly

2. Has the statistical analysis been performed appropriately and rigorously? 

Reviewer #1: Yes

Reviewer #2: No

3. Have the authors made all data underlying the findings in their manuscript fully available?

Reviewer #1: Yes

Reviewer #2: Yes

4. Is the manuscript presented in an intelligible fashion and written in standard English?

Reviewer #1: Yes

Reviewer #2: No

5. Review Comments to the Author

Reviewer #1: The manuscript prepared in good manner. However, it requires some corrections to improve the quality. For instance, it is very important to provide brief information about weed problems at the beginning of the introduction section

For other suggestions, see the comments and suggestions in the attached MS file

Reviewer #2: Manuscript No: PONE-D-24-15577

Manuscript Title: The Relative Importance of Herbicide Use for Conservation Tillage Adoption by U.S. Corn and Soybean Producers.

A major revision is being suggested for the submitted manuscript. Following are the point-to-point comments that need to be addressed for further review:

1. The abstract needs to be rewritten, and the proper motivation followed by clear objectives and major findings with quantitative support need to be provided in the revised abstract.

2. The introduction needs to be revised, and the research gap is not clear. Hence, authors are advised to rewrite/ update the introduction to clearly present the research gap, and in continuing with that, proper objectives must be formulated.

3. Authors must avoid using the first person in the technical writing process (avoid the use of i/we//our/us, etc.)

4. All the headings and subheadings must be properly numbered starting from the introduction with no 1.

5. The explanation with the figures caption can be kept as text in the main body of the manuscript and can be discussed properly.

6. The results and analysis must be presented and discussed in appropriate sections. Here, the authors have presented and analyzed the results in the prior sections. Hence, it is suggested that the authors follow the IMRAD paper structure for better presentation and understanding.

7. The analysis of the obtained results is very poor, and there is no comparison with any existing/ published work. Hence, the authors are suggested to update the results, compare them in the discussion section, and provide appropriate reasoning for the improved behavior in the proposed method.

8. Conclusions must be rewritten (in max 200 words), and they must convey the major findings of the research with quantitative support.

I wish the author good luck.

6. PLOS authors have the option to publish the peer review history of their article (what does this mean?). If published, this will include your full peer review and any attached files.

Reviewer #1: No

Reviewer #2: **Yes: **Dr. Narendra Khatri

---

## [Author Response · Author response to Decision Letter 0]

17 Sep 2024

Please see our response to reviewer comments document attached.

---

## [Editor Report · Decision Letter 1]

30 Sep 2024

The Relative Importance of Herbicide Use for Conservation Tillage Adoption by U.S. Corn and Soybean Producers

PONE-D-24-09492R1

Dear Dr. Nemec Boehm,

We’re pleased to inform you that your manuscript has been judged scientifically suitable for publication and will be formally accepted for publication once it meets all outstanding technical requirements.

Kind regards,

Dafeng Hui, Ph.D.

Academic Editor

PLOS ONE

Additional Editor Comments (optional):

The authors have made great efforts and adequately addressed these concerns.
---

## [Editor Report · Acceptance letter]

30 Oct 2024

PONE-D-24-09492R1 

PLOS ONE

Dear Dr. Nemec Boehm, 

I'm pleased to inform you that your manuscript has been deemed suitable for publication in PLOS ONE. Congratulations! Your manuscript is now being handed over to our production team.

Kind regards, 

on behalf of

Dr. Dafeng Hui 

Academic Editor

PLOS ONE